# Continual Learning via Continual Weighted Sparsity and Meta-Plasticity Scheduling

## Abstract

Continual Learning (CL) is fundamentally challenged by the stability-plasticity dilemma: the trade-off between acquiring new information and maintaining past knowledge. To address the stability, many methods keep a replay buffer containing a small set of samples from prior tasks and employ parameter isolation strategies that allocate separate parameter subspaces for each task, reducing interference between tasks. To get more refined, task-specific groups, we adapt a dynamic sparse training technique and introduce a continual weight score function to guide the iterative pruning process over multiple rounds of training. We refer to this method as the continual weighted sparsity scheduler. Furthermore, with more incremental tasks introduced, the network inevitably becomes saturated, leading to a loss of plasticity, where the model's adaptability decreases due to dormant or saturated neurons. To mitigate this, we draw inspiration from biological meta-plasticity mechanisms, and develop a meta-plasticity scheduler to dynamically adjust these task-specific groups' learning rates based on the sensitive score function we designed, ensuring a balance between retaining old knowledge and acquiring new skills. The results of comparison on popular datasets demonstrate that our approach consistently outperforms existing state-of-the-art methods, confirming its effectiveness in managing the stability-plasticity trade-off.

## 1 Introduction

To navigate the complexities of real-world environments, an intelligent system must continuously learn, adapt, and apply knowledge over time (Parisi et al., 2019; Kudithipudi et al., 2022). This need has driven the study of continual learning (CL), where a typical setting is to learn a sequence of tasks incrementally while retaining performance on previous tasks, despite not having access to all tasks simultaneously. These tasks may involve acquiring new skills, revisiting previously learned ones, or adapting to different environments and contexts, each posing its own set of challenges (Hadsell et al., 2020; Wang et al., 2024a).

Unlike traditional machine learning models, which assume a static data distribution, CL involves learning from dynamic data distributions across a sequence of tasks. A key challenge in CL is the *stability-plasticity dilemma* (Grossberg, 1987), which arises when balancing the need to acquire new knowledge while preserving past knowledge. Stability is threatened when learning new tasks causes the model to overwrite or degrade the representations learned from previous tasks, particularly at task boundaries where shifts in data distribution are most pronounced (Robins, 1995; Buzzega et al., 2020). This can result in a sharp performance decline on older tasks, or in extreme cases, complete forgetting of previously acquired knowledge (Parisi et al., 2019). On the other hand, maintaining plasticity is crucial for adapting to new tasks and incorporating fresh information, but excessive plasticity can erode previously learned skills. Achieving a suitable trade-off between stability and plasticity is essential, yet remains a fundamental challenge for CL algorithms.

Existing CL algorithms typically retain a small buffer of samples from previous tasks during the training of new tasks, which helps mitigate the distribution shift and preserve stability by maintaining past knowledge (Verwimp et al., 2021; Bhat et al., 2022). Building on this common strategy, two primary approaches have been proposed to address the stability challenge: replay-based methods and parameter isolation methods. Replay-based methods optimize the use or selection of memory buffers, while parameter isolation methods allocate separate parameter subspaces for each task,

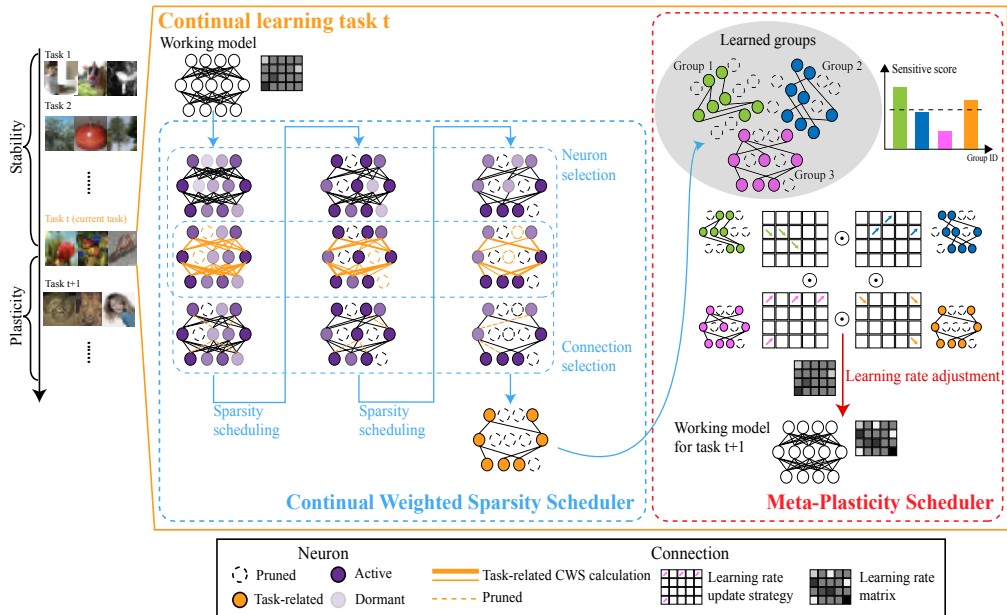

Figure 1: Our continual learning process is divided into two main steps: (1) using the continual weighted sparsity scheduler to identify task-specific neuron groups, involving iteratively pruning neurons and connections, and (2) using the meta-plasticity scheduler to adjust learning rate for each connection based on the sensitive score for each group. In the continual weighted sparsity scheduler, the intensity of the purple color for each neuron indicates its activation value, with darker shades corresponding to higher activations. During the neuron selection step (from the top to the middle row), neurons with lower activation values are pruned. Additionally, the width of the orange connections represents the continual weighted score (CWS). Connections with lower scores are pruned during the connection selection step (from the middle to the bottom row). In the meta-plasticity scheduler, each group has a different learning rate update strategy based on its sensitive score. Ultimately, the entire model updates the learning rates for all connections, stored in a learning rate matrix. Lighter colors indicate higher learning rates.

reducing interference between tasks (Wang et al., 2024a). In this work, we mainly focus on the parameter isolation approaches. Previous work typically relies on a fixed pruning strategy for each task, applying a one-time pruning with a predefined sparsity based on a score function (Mallya & Lazebnik, 2018; Vijayan et al., 2023). To improve upon this, we propose the *continual weighted sparsity scheduler*, inspired by recent dynamic sparse training techniques. Specifically, instead of applying a single round of pruning, our method iteratively prunes the network with a gradually increasing sparsity over multiple rounds of training. This ensures that the most active neurons and their corresponding connections, which are most relevant to the current task, are retained. The iterative pruning process thus results in a more refined, task-specific neuron and connection group, preserving knowledge more effectively.

Since the network capacity is limited, as more incremental tasks are introduced, the network will eventually become saturated. Recent studies have demonstrated that neural networks may gradually lose their capacity to learn from new experiences, a phenomenon referred to as the *loss of plasticity*, which is potentially caused by dormant or saturated neurons, further complicating the learning process (Lyle et al., 2023; Sokar et al., 2023). To address this issue, we adopt a mechanism inspired by biological systems known as *meta-plasticity* (Kudithipudi et al., 2022), which refers to the phenomenon where the strength of individual synapses can be modulated by neural activity, with the ease of synaptic strengthening or weakening varying over time. This is also described as the "plasticity of plasticity", meaning that a synapse's capacity for change depends on its internal biochemical state. These states are influenced by the synapse's history of modifications and recent neural activity, enabling fast learning and slow forgetting (Abraham & Bear, 1996; Abraham, 2008). Building on this concept, we propose the *meta-plasticity scheduler*. After identifying task-specific

neuron and connection groups during training, we calculate a sensitivity score for each group by measuring the average normalized magnitude difference across all connections between the two most recent tasks. During subsequent model updates, the learning rate of each connection is dynamically adjusted based on the sensitive scores within these groups. Unlike previous approaches that reset connections of dormant neurons through weight reinitialization, our method provides a more fine-grained, connection-level, and task-aware adjustment, allowing for a flexible and dynamic tuning of connections. By considering the influence of previously learned knowledge on the current task, our approach ensures that the network maintains better plasticity in the CL setting, facilitating both knowledge retention and adaptation to new tasks.

In summary, to address the stability-plasticity dilemma in CL, we propose a framework that integrates the continual weighted sparsity scheduler and the meta-plasticity scheduler. To validate our approach, we comprehensively compare it against state-of-the-art CL methods on popular datasets. We also evaluate the stability and plasticity of our models over a long sequence of tasks, providing deeper insights into the effectiveness of our method. Comprehensive validation tests and analyses consistently demonstrate that our framework outperforms existing approaches, effectively addressing the stability-plasticity trade-off in CL.

## 2 Related Work

**Approaches to address stability in CL.** To address stability in CL, various approaches aim to prevent or minimize this degradation, ensuring that the network retains knowledge from previous tasks even as it learns new ones. One prevalent strategy involves storing a limited number of past training samples in a small memory buffer (Ratcliff, 1990; Robins, 1995), similar to the experience replay mechanism observed in the brain (Rasch & Born, 2007). Based on this consensus, researchers have developed two primary approaches to further tackle the stability issue: replay-based approaches and parameter-isolation approaches. Replay-based approaches focus on optimizing both buffer construction and buffer exploitation to make better use of the limited memory buffer, enhancing the retention of past knowledge. GCR (Tiwari et al., 2022) introduces a selection mechanism that approximates the gradients of previously seen data to update the buffer. DER++ (Buzzega et al., 2020) and CLS-ER (Arani et al., 2022) enhance consistency in predictions by using both soft targets and ground-truth labels. MRFA (Zheng et al., 2024) refines decision boundaries by augmenting the block-level features of rehearsal samples across multiple layers. On the other hand, parameter-isolation approaches have explored task-specific parameter isolation methods to further minimize interference between tasks. For example, PackNet (Mallya & Lazebnik, 2018), CLNP (Golkar et al., 2019), SparCL Wang et al. (2022a) and NISPA (Gurbuz & Dovrolis, 2022) leverage the over-parameterization of deep neural networks (DNNs) to accommodate multiple tasks within a fixed model capacity. Inspired by the brain, they jointly learn both the connection strengths and sparse task-specific subnetworks, effectively isolating parameters across tasks. More recently, TriRE (Vijayan et al., 2023) introduces a method for retaining the most prominent neurons while promoting the activation of less active ones, and TPL (Lin et al., 2024) proposes a more principled approach for task-ID prediction to enhance task isolation. Though effective, these methods typically use a fixed pruning strategy with a predefined sparsity, leading to less accurate task-specific sub-networks and reduced downstream performance. In contrast, our continual weighted sparsity scheduler employs iterative pruning, progressively increasing the sparsity across multiple training rounds. This gradually refines the network, preserving key neurons and connections. Experiments show our method, as a novel parameter isolation technique, outperforms existing replay-based and parameter isolation approaches in retaining task-specific knowledge, thus better addressing the stability challenge in CL.

**Approaches to maintain plasticity in CL.** To address the challenge of plasticity in CL, several strategies have been proposed, most of which are based on reinitializing some or all of the network's weights during training. For instance, Zhou et al. (2021) suggest that selective forgetting can enhance generalization, while Zhang et al. (2022) demonstrate that resetting different layers has varying impacts on network performance. Additionally, Zhao et al. (2023) introduced a method to fine-tune task-specific parameters on buffered data to improve plasticity. Refresh (Wang et al., 2024b) dynamically eliminates outdated information by refreshing some of the old task-specific weights from the CL model, thereby enhancing the retention of older knowledge while efficiently acquiring information for new tasks. Unlike these previous approaches, we leverage the fundamen-

tal mechanism of meta-plasticity found in biological systems (Langille & Brown, 2018). Instead of directly reinitializing weights of the model, our meta-plasticity scheduler dynamically adjusts the ease with which neurons adapt, depending on their activity levels on recent tasks. This mechanism enables a more nuanced and adaptive regulation of neural plasticity, allowing for greater flexibility and precision in controlling how learning unfolds in the network.

## 3 METHODS

We begin by outlining the definitions and preliminaries of CL in Section 3.1, followed by an overview of our system in Section 3.2. We then introduce our proposed continual weighted sparsity scheduler in Section 3.3, and the meta-plasticity scheduler in Section 3.4.

### 3.1 PRELIMINARIES

CL is characterized by learning from dynamic data distributions. In practice, training samples of different distributions arrive sequentially. A working model $f_\theta$ parameterized by $\theta$ needs to learn corresponding task(s) with limited or no access to previous training samples and perform well on their test sets. Formally, CL problems typically comprise $t \in \{1, 2, \ldots, T\}$ sequential tasks, with $c$ classes per task, and data that appear incrementally over time. Each task has an associated task-specific data distribution: $(x_t, y_t) \in D_t$, where $x_t$ is the input data, $y_t$ is the data label, and $t$ is the task identity. The overall objective of CL is to maintain performance on previous datasets $D_i$ where $i \in \{1, 2, \ldots, t-1\}$, while ensuring sufficiently good performance on the current dataset $D_t$. In this work, we consider two well-known CL scenarios, class-incremental learning (Class-IL) and task-incremental learning (Task-IL), both of which have disjoint label spaces across tasks. In the former, task identities are provided only during training, whereas in the latter, task identities are available during both training and testing.

Similar to common approaches, we maintain a memory buffer $D_m$ to retain information from previous tasks. Considering the constraints of CL, the model does not have infinite storage for previous experience, and thus $|D_m| \ll |D_t|$. Given the current task data $D_t$ and the memory buffer $D_m$, a combination of the task-wise loss $\mathcal{L}_t$ and the experience replay-based loss $\mathcal{L}_{\text{rep}}$ is commonly used during the training of the working model $f_\theta$:

$$\begin{cases} \mathcal{L}_t = \mathbb{E}_{(x_i, y_i) \sim D_t}[\mathcal{L}_{\text{ce}}(f_\theta(x_i), y_i)] \\ \mathcal{L}_{\text{rep}} = \mathbb{E}_{(x_j, y_j) \sim D_m}[\mathcal{L}_{\text{ce}}(f_\theta(x_j), y_j)] \end{cases}, \tag{1}$$

where $\mathcal{L}_{\text{ce}}$ is the cross-entropy loss. $\mathcal{L}_t$ is computed on the current task data $D_t$ to promote plasticity, while $\mathcal{L}_{\text{rep}}$ is computed from the memory buffer $D_m$ to enhance stability.

### 3.2 OVERVIEW OF OUR SYSTEM

As shown in Figure 1, our system contains two main steps that alternate continuously during the task training process: (1) filter out task-specific neuron groups that are highly active to the current task, and then integrate them into the existing neuron group pool; (2) update the meta-plasticity of all groups based on their sensitive scores.

Specifically, for a new task $t$, we perform multiple rounds of network pruning by gradually increasing target sparsity and iteratively pruning neurons and connections in the working model. This iterative pruning process refines a more task-specific group, preserving knowledge more effectively. The refined group is subsequently integrated into the existing pool of neuron groups. Once the task-specific neuron groups are identified, we calculate the sensitive score for each group and adjust the learning rates of connections within those groups based on their scores. This adjustment either releases or suppresses the neuron update capacity, achieving an optimal balance between stability and plasticity. Finally, we employ reservoir sampling to update the replay buffer $D_m$ and reinitialize the most dormant neurons for future tasks. The entire process is detailed in Algorithm 1 (Appendix B.1).

### 3.3 CONTINUAL WEIGHTED SPARSITY SCHEDULER

To preserve task-specific information and address the stability challenge, we propose the continual weighted sparsity scheduler, inspired by recent dynamic sparse training techniques, to enhance the

selection of parameters when performing parameter isolation in CL. Specifically, for the current task $t$, we iteratively perform multiple rounds of network pruning. At the beginning of each pruning round, we calculate the target sparsity which is raised from the previous round (Step 1). Next, we utilize the activation score of neurons to perform neuron selection based on the target sparsity (Step 2a). Then, for the selected neurons, we apply our proposed *continual weighted score* (CWS) function to further refine the selection of connections (Step 2b). The continual weighted sparsity scheduler allows us to progressively obtain a network with increasing sparsity, ultimately reaching the pre-defined target sparsity. Throughout the rounds, information from task $t$ is maximally preserved.

**Step 1. Sparsity scheduling.**  For a new task $t$, we first calculate the available network sparsity that has not been allocated to previous tasks, denoted as $S_{t-1}$. Then, we assign a fixed sparsity $\Delta S$ to task $t$, resulting in the target sparsity $S_{t,N}$ after training the task for $N$ epochs.

Denote the working model $f_\theta$ as a graph $g = (\mathcal{N}, \mathcal{E})$, where $\mathcal{N}$ is the set of neurons in the model and $\mathcal{E} \subseteq \mathcal{N} \times \mathcal{N}$ is the set of connections between the neurons. We aim to decompose $g$ into $T$ task-specific sub-networks. For task $t$, the corresponding sub-network is denoted as $g_t = (\mathcal{N}_t, \mathcal{E}_t)$, where $\mathcal{N}_t \subseteq \mathcal{N}$ and $\mathcal{E}_t \subseteq \mathcal{N}_t \times \mathcal{N}_t$. Then we have:

$$S_t = 1 - \frac{|\bigcup_{i=1}^{t} \mathcal{N}_i|}{|\mathcal{N}|}, \tag{2}$$

with $S_0 = 100\%$. As mentioned above, each task is allocated a pre-defined sparsity $\Delta S$, meaning that $\frac{|\mathcal{N}_t|}{|\mathcal{N}|} = \Delta S$. It is important to note that $S_t$ may not be equal to $1 - t \times \Delta S$ because there may be overlapping neurons and connections between these sub-networks. Here, we adopt an automated gradual pruning algorithm (Zhu & Gupta, 2017) to achieve task-wise sparsity scheduling. We first set the target sparsity of the model after training total $N$ epochs as:

$$S_{t,N} = \max(0, S_{t-1} - \Delta S). \tag{3}$$

In our experiments, we set $\Delta S$ to 15%, as this level of sparsity has shown comparable performance to that of a fully dense network (Han et al., 2015; Graesser et al., 2022). Then, during the multi-round training process of the task $t$, we use the following sparsity scheduling:

$$S_{t,n} = S_{t,N} - S_{t,N}(1 - \frac{n}{N})^3, \qquad n = 1, 2, \ldots, N, \tag{4}$$

where $S_{t,n}$ is the sparsity of $f_\theta$ after training $n$ epochs. Next, we distribute the overall sparsity $S_{t,n}$ to the target sparsity $S_{t,n}^{(l)}$ for each layer $l$ based on the number of neurons $d^{(l)}$ in each layer, guiding the selection of the most active neurons. We adopt the *Erdős-Rényi* method (Mocanu et al., 2018) here, and the sparsity distribution across layers is provided in more detail in Appendix B.2.

**Step 2a. Neuron selection.**  Once we obtain $S_{t,n}^{(l)}$, we need to prune the layer by selecting essential neurons first. Let $a_i^{(l)}(x)$ denote the activation of neuron $i$ in layer $l$ under input $x$ from a batch of training data $\mathcal{B}_t \subset D_t$. Then we define the activation score of a neuron $i$ in layer $l$ via the normalized average of its activation as follows:

$$A_i^{(l)} = \frac{\mathbb{E}_{x \in \mathcal{B}_t} |a_i^{(l)}(x)|}{\sum_{k=0}^{d^{(l)}} \mathbb{E}_{x \in \mathcal{B}_t} |a_k^{(l)}(x)|}. \tag{5}$$

Neurons with high activation scores within the top $S_{t,n}^{(l)}$ will be selected.

**Step 2b. Connection selection.**  After selecting the most active neurons, we select the most important connections between these neurons based on our continual weighted score (CWS) function, which extends the continual weight importance proposed by Wang et al. (2022a):

$$\text{CWS}(\omega) = \|\omega\|_1 + \alpha_1(\|\frac{\partial \hat{\mathcal{L}}_{ce}(D_t; \theta)}{\partial \omega}\|_1 + \|\frac{\partial \hat{\mathcal{L}}_{\text{new}}(D_t; \theta)}{\partial \omega}\|_1) + \alpha_2\|\frac{\partial \hat{\mathcal{L}}_{ce}(D_m; \theta)}{\partial \omega}\|_1, \tag{6}$$

where $\omega \in \theta$ is the weight, $\hat{\mathcal{L}}_{ce}(D_t; \theta)$ denotes the single-head form of the cross-entropy loss on the current task data $D_t$, which only takes into account the classes relevant to the current task by

masking out the logits of other classes, $\hat{\mathcal{L}}_{ce}(D_m; \theta)$ denotes the loss on the memory buffer data $D_m$. Compared to Wang et al. (2022a), we introduce the task-aware term $\hat{\mathcal{L}}_{new}(D_t; \theta)$ to improve the model's ability to recognize task boundaries, which is the cross-entropy loss for new/old task distinction. The CWS ensures that we maintain: (1) weights of greater magnitude for output stability, (2) weights significant for the current task for learning capacity, (3) weights significant for task distinction and (4) weights significant for previous tasks to prevent catastrophic forgetting, with two hyper-parameter $\alpha_1$ and $\alpha_2$ are used to regulate the weight of current and buffered data, respectively. In this paper, we follow Wang et al. (2022a) and set $\alpha_1 = 0.5$ and $\alpha_2 = 1$.

After iterative pruning of the neurons and connections, we obtain a group of neurons and connections $g_t$ for task $t$. As $T$ tasks are sequentially introduced, finally, we will get a set of groups of neurons and connections, denoted as $\mathcal{G} = \{g_1, g_2, \ldots, g_T\}$.

### 3.4 META-PLASTICITY SCHEDULER

A common approach to achieving task isolation in CL is to freeze task-specific parameters once the task is completed (Mallya & Lazebnik, 2018; Vijayan et al., 2023). While this strategy helps preserve acquired knowledge, it limits the network's ability to adapt to new tasks and challenges. In contrast, biological systems utilize meta-plasticity, a mechanism where synapses dynamically adjust their capacity to change based on their modification history. This concept is crucial for enhancing a network's long-term learning potential and adaptability (Kudithipudi et al., 2022).

Inspired by that, we propose a neuro-level dynamic learning rate schedule strategy. Each neuron has an independent learning rate schedule strategy based on its sensitivity to recent activities. This approach suppresses overly active parts to reduce the forgetting of old knowledge while simultaneously identifying and revitalizing gradually rigid sections, thereby maintaining the plasticity.

Specifically, we first calculate the normalized magnitude difference of the weight for connection $e$ between layer $l$ and layer $l + 1$ after training two consecutive tasks, as follows:

$$C_e = \frac{\|\omega_t^e - \omega_{t-1}^e\|_1}{\|W_t^{(l)} - W_{t-1}^{(l)}\|_1}, \tag{7}$$

where $\omega_t^e \in \theta$ denotes the weight of the connection $e$ learned after training on task $t$, and $W^{(l)} \in \theta$ is the weight matrix between layer $l$ and layer $l + 1$.

Then we measure the sensitivity of all groups based on the average normalized magnitude difference across all connections within each group. Given a set of groups of neurons and connections $\mathcal{G}$ from Section 3.3, we define a sensitive score $\mathrm{SS}_{g_t}$ for each group $g_t$:

$$\mathrm{SS}_{g_t} = \frac{(1/|\mathcal{E}_t|) \sum_{e \in \mathcal{E}_t} C_e}{\sum_{k=1}^{|\mathcal{G}|} ((1/|\mathcal{E}_k|) \sum_{e \in \mathcal{E}_k} C_e)} \times |\mathcal{G}|, \tag{8}$$

where $\mathcal{E}_t$ is the connections in $g_t$. The learning rates of connections within $g_t$ are then updated as:

$$\mathrm{lr}_{g_t} \leftarrow \mathrm{lr}_{g_t} \times \lambda^{(1 - \mathrm{SS}_{g_t})}, \tag{9}$$

where $\lambda > 1$ is used to control the magnitude of change of the learning rate. Based on the group update strategy, we have the update strategy for each connection to achieve meta-plasticity scheduling:

$$\mathrm{lr}_e \leftarrow \mathrm{lr}_e \times \prod_{\substack{g_t \in \mathcal{G} \\ e \in \mathcal{E}_t}} \lambda^{(1 - \mathrm{SS}_{g_t})}. \tag{10}$$

Here, we consider groups with an $\mathrm{SS} < 1$ to be relatively inactive, as their parameter variation is smaller than the average across all groups. For these groups, we increase their meta-plasticity, while for those with an $\mathrm{SS} > 1$, we do the opposite. We also note that when $\lambda = 0$, it is equal to the strategy of freezing task-specific parameters.

## 4 EXPERIMENTS

### 4.1 EXPERIMENTAL SETTINGS

**Datasets.** To evaluate the performance of our method in Class-IL and Task-IL scenarios, we follow standard image classification benchmarks in CL (Rebuffi et al., 2017; Wu et al., 2019) and employ

three different datasets: CIFAR-10, CIFAR-100, and Tiny-ImageNet. Specifically, CIFAR-10 is divided into 5 disjoint tasks with 2 classes per task. CIFAR-100 is divided into 10 tasks with each containing 10 disjoint classes. Tiny-ImageNet consists of 200 classes, divided into 10 tasks with 20 classes per task. The statistics of the different datasets are provided in Appendix A.1.

**Baselines.** We extensively compare our method with representative baselines, including replay-based approaches: ER (Chaudhry et al., 2019), DER++ (Buzzega et al., 2020), CLS-ER (Arani et al., 2022), ER-ACE (Caccia et al., 2021), Co$^2$L (Cha et al., 2021), GCR (Tiwari et al., 2022), Refresh (Wang et al., 2024b), MRFA (Zheng et al., 2024) and task-isolation approaches: SparCL Wang et al. (2022a), TriRE (Vijayan et al., 2023), TPL (Lin et al., 2024). Additionally, following previous CL work, we report a lower bound (SGD without support) and an upper bound (Joint training on the complete dataset).

**Metrics.** Overall performance is primarily evaluated by average accuracy (AA). Let $a_{k,j} \in [0, 1]$ denote the classification accuracy evaluated on the test set of the $j$-th task after incremental learning of the $k$-th task ($j \leq k$). AA is computed as $\frac{1}{T} \sum_{j=1}^{T} a_{T,j}$ after learning a total of $T$ tasks. Additionally, following Sarfraz et al. (2022), we evaluate the model's stability, plasticity, and the trade-off between the two; the details of how these three metrics are calculated can be found in Appendix C.1. For each experiment, we fix the order of the classes and report the average AA and one standard deviation across all tasks over 5 runs with different initializations.

Table 1: Comparison of the overall performance of prior methods across various CL scenarios.

| Methods | CIFAR-10 | | CIFAR-100 | | Tiny-ImageNet | |
|---|---|---|---|---|---|---|
| | Class-IL | Task-IL | Class-IL | Task-IL | Class-IL | Task-IL |
| SGD | $19.62_{\pm0.05}$ | $61.02_{\pm3.33}$ | $17.49_{\pm0.28}$ | $40.46_{\pm0.99}$ | $7.92_{\pm0.26}$ | $18.31_{\pm0.68}$ |
| Joint | $92.20_{\pm0.15}$ | $98.31_{\pm0.12}$ | $70.56_{\pm0.28}$ | $86.19_{\pm0.43}$ | $59.99_{\pm0.19}$ | $82.04_{\pm0.10}$ |
| ER | $44.79_{\pm1.86}$ | $91.19_{\pm0.94}$ | $21.40_{\pm0.22}$ | $61.36_{\pm0.35}$ | $8.57_{\pm0.04}$ | $38.17_{\pm2.00}$ |
| DER++ | $64.88_{\pm1.17}$ | $91.92_{\pm0.60}$ | $29.60_{\pm1.14}$ | $62.49_{\pm1.02}$ | $10.96_{\pm1.17}$ | $40.87_{\pm1.16}$ |
| CLS-ER | $61.88_{\pm2.43}$ | $93.59_{\pm0.87}$ | $43.38_{\pm1.06}$ | $72.01_{\pm0.97}$ | $17.68_{\pm1.65}$ | $52.60_{\pm1.56}$ |
| ER-ACE | $62.08_{\pm1.44}$ | $92.20_{\pm0.57}$ | $35.17_{\pm1.17}$ | $63.09_{\pm1.23}$ | $11.25_{\pm0.54}$ | $44.17_{\pm1.02}$ |
| Co$^2$L | $65.57_{\pm1.37}$ | $93.43_{\pm0.78}$ | $31.90_{\pm0.38}$ | $55.02_{\pm0.36}$ | $13.88_{\pm0.40}$ | $42.37_{\pm0.74}$ |
| GCR | $64.84_{\pm1.63}$ | $90.80_{\pm1.05}$ | $33.69_{\pm1.40}$ | $64.24_{\pm0.83}$ | $13.05_{\pm0.91}$ | $42.11_{\pm1.01}$ |
| SparCL | $66.30_{\pm0.98}$ | $94.06_{\pm0.45}$ | $38.49_{\pm0.76}$ | $68.42_{\pm0.55}$ | $13.25_{\pm0.52}$ | $43.21_{\pm0.58}$ |
| TriRE | $68.17_{\pm0.33}$ | $92.45_{\pm0.18}$ | $\mathbf{43.91}_{\pm0.18}$ | $71.66_{\pm0.44}$ | $20.14_{\pm0.19}$ | $55.95_{\pm0.78}$ |
| TPL | $70.06_{\pm0.47}$ | $92.33_{\pm0.32}$ | $36.90_{\pm0.42}$ | $76.53_{\pm0.27}$ | $20.06_{\pm0.77}$ | $54.20_{\pm0.51}$ |
| Refresh | $74.42_{\pm0.82}$ | $94.64_{\pm0.38}$ | $38.49_{\pm0.76}$ | $77.71_{\pm0.85}$ | $20.81_{\pm1.28}$ | $54.06_{\pm0.79}$ |
| MRFA | $73.38_{\pm0.54}$ | $93.44_{\pm0.16}$ | $37.23_{\pm0.65}$ | $75.83_{\pm0.48}$ | $21.68_{\pm0.55}$ | $54.59_{\pm0.42}$ |
| **Ours** | $\mathbf{75.31}_{\pm0.71}$ | $\mathbf{95.79}_{\pm0.65}$ | $40.61_{\pm0.58}$ | $\mathbf{79.91}_{\pm0.63}$ | $\mathbf{23.25}_{\pm0.59}$ | $\mathbf{58.32}_{\pm0.73}$ |

## 4.2 EXPERIMENTAL RESULTS

**Overall performance.** As shown in Table 1, our method consistently outperforms the baselines across most datasets in both Class-IL and Task-IL settings. Notably, as the dataset complexity and the number of tasks increase from CIFAR-10 to Tiny-ImageNet, the performance gap between our method and the baselines grows considerably. More results on the larger ImageNet-1K dataset, along with comparisons against additional baselines, are provided in Appendix C.3.

**Stability-Plasticity trade-off.** We further analyze the trade-off between stability and plasticity achieved by our method, as well as the performance across all tasks after training. From Figure 2a, it is evident that our method demonstrates the best stability while maintaining near-optimal plasticity, which leads to the most favorable stability-plasticity trade-off. This explains why our approach achieves the best overall performance. Figure 2b provides additional insight, showing that our method significantly outperforms others on the earlier tasks. We believe this is due to the task isolation mechanism, which contributes to the superior stability of our method compared to others. However, when looking at the last four tasks, we observe a slight performance decline, especially on the final task, where the performance is not the best. This may explain why our plasticity is

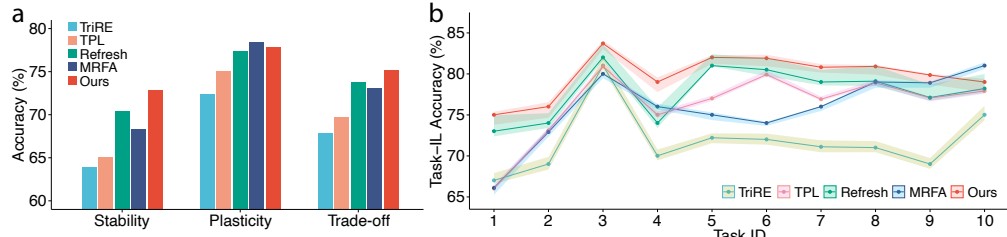

Figure 2: **a**. Stability-Plasticity trade-off for CL models trained on CIFAR-100 with 10 tasks. **b**. Comparison of our method against other representative baselines in terms of Task-IL accuracy on the CIFAR-100 dataset divided into 10 tasks. We report the average accuracy over five runs with different seeds; the shaded area indicates the range between the minimum and maximum values.

not the highest. We suspect this is due to the network gradually becoming saturated, leaving insufficient neurons available for learning new tasks. More results for Tiny-ImageNet are provided in Appendix C.3.

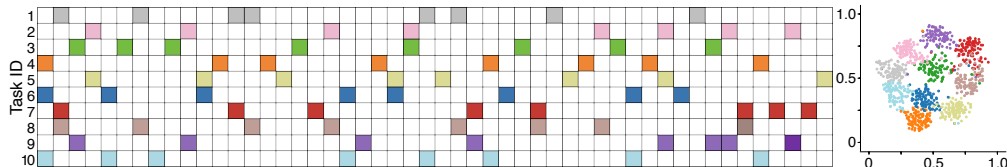

Figure 3: **Left**: Visualization of the neuron groups extracted for each task for the last shortcut layer when training on CIFAR-100 with 10 tasks. Each row from top to bottom represents a task, from task 1 to task 10. **Right:** Visualization of the feature vectors from the last convolutional layer using t-SNE, with different colors representing different tasks, and the colors match those in the left one.

**Task isolation.** To validate the effectiveness of task isolation in our method, we analyze the extracted neuron groups, with the results from the last shortcut layer shown on the left of Figure 3. Each row represents the neuron group extracted for a specific task. From the first few rows, we can see that the neurons allocated to each task typically have no overlap, confirming the effectiveness of our approach in minimizing interference between tasks through parameter isolation. However, as the number of tasks increases and the network reaches saturation, neurons used by older tasks are gradually released. This results in some overlap between the neurons used for later tasks and those for earlier tasks, as seen in the lower rows. Despite this, the overlap between adjacent tasks remains well-controlled. While we aim for complete task separation, the overlap between neuron groups in rows suggests the similarity between tasks. We also visualize the features of task samples using t-SNE, as shown on the right side of Figure 3. The visualization reveals good separability between different tasks, though there is some overlap at the boundaries of certain tasks. For example, tasks 7 and 8, represented by the red and brown clusters, exhibit more overlap, which can also be observed in the left-side neuron visualization, where these tasks share more neurons compared to others. We believe this overlap is due to inherent similarities between the tasks themselves.

Table 2: Comparison of the overall performance of prior methods with 20 tasks.

| Methods | CIFAR-100 | | Tiny-ImageNet | |
|---|---|---|---|---|
| | Class-IL | Task-IL | Class-IL | Task-IL |
| SGD | $18.91_{\pm0.34}$ | $45.31_{\pm0.76}$ | $10.47_{\pm0.47}$ | $23.22_{\pm0.52}$ |
| Joint | $74.12_{\pm0.42}$ | $89.81_{\pm0.58}$ | $66.37_{\pm0.21}$ | $86.94_{\pm0.23}$ |
| TriRE | $38.29_{\pm0.66}$ | $76.62_{\pm0.37}$ | $27.41_{\pm0.79}$ | $55.87_{\pm0.44}$ |
| TPL | $37.38_{\pm0.94}$ | $77.64_{\pm0.55}$ | $26.85_{\pm0.86}$ | $54.99_{\pm0.75}$ |
| Refresh | $39.53_{\pm0.85}$ | $79.81_{\pm0.32}$ | $27.59_{\pm0.64}$ | $55.52_{\pm0.51}$ |
| MRFA | $38.52_{\pm0.63}$ | $78.93_{\pm0.72}$ | $27.72_{\pm0.65}$ | $56.82_{\pm0.52}$ |
| **Ours** | $\mathbf{41.69_{\pm0.57}}$ | $\mathbf{82.46_{\pm0.61}}$ | $\mathbf{30.53_{\pm0.66}}$ | $\mathbf{59.82_{\pm0.81}}$ |

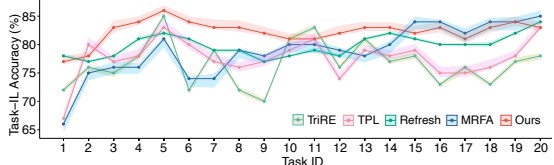

Figure 4: Comparison on the CIFAR-100 dataset with 20 tasks. We report the average accuracy over five different seeds; the shaded area indicates the range between the minimum and maximum values.

**Performance on long sequences of tasks.** As mentioned earlier, when the number of tasks increases, network saturation may occur, potentially affecting performance. To evaluate this, we conduct experiments on a longer task sequence, with the results shown in Table 2. Our method consistently outperforms all baselines in both Class-IL and Task-IL scenarios. We report the performance of all 20 tasks after training, as illustrated in Figure 4. Similar to the case with 10 tasks, our method demonstrates superior performance in preserving the accuracy of the earlier tasks. Furthermore, it maintains relatively high accuracy and exhibits less fluctuation for newly added tasks compared to other methods. More results on Tiny-ImageNet and ImageNet-1K can be found in Appendix C.3.

## 4.3 ABLATION STUDY

**Continual weighted sparsity scheduler.** To demonstrate the advantages of our approach, we compare our continual weighted sparsity scheduler with two baselines: (1) Static: a network trained with fixed sparsity from scratch, and (2) RigL (Evci et al., 2020): the foundation of our method, which uses a dynamic sparse training approach that prunes connections based solely on the magnitude of weights, meaning that only the first term in Equation 6 is used to compute the continual weighted score. As shown in Table 3, the dynamic sparsity approach outperforms static sparse training, and our continual weighted sparsity scheduler yields even more promising results.

Table 3: Comparison of different sparse training methods across various CL scenarios with 10 tasks.

| Methods | CIFAR-100 | | Tiny-ImageNet | |
|---|---|---|---|---|
| | Class-IL | Task-IL | Class-IL | Task-IL |
| Static | $37.45_{\pm 0.56}$ | $76.39_{\pm 0.25}$ | $21.36_{\pm 0.41}$ | $54.28_{\pm 0.57}$ |
| RigL | $39.49_{\pm 0.81}$ | $77.78_{\pm 0.42}$ | $22.54_{\pm 0.64}$ | $56.75_{\pm 0.55}$ |
| **Ours** | $\mathbf{40.61_{\pm 0.58}}$ | $\mathbf{79.91_{\pm 0.63}}$ | $\mathbf{23.25_{\pm 0.59}}$ | $\mathbf{58.32_{\pm 0.73}}$ |

Table 4: The average accuracy for different $\lambda$ in Equation 10 across various CL scenarios.

| $\lambda$ | CIFAR-100 | | Tiny-ImageNet | |
|---|---|---|---|---|
| | Class-IL | Task-IL | Class-IL | Task-IL |
| 0 | $38.14_{\pm 0.77}$ | $75.26_{\pm 0.45}$ | $21.81_{\pm 0.59}$ | $54.84_{\pm 0.84}$ |
| 1 | $34.70_{\pm 0.35}$ | $70.96_{\pm 0.65}$ | $17.20_{\pm 0.72}$ | $49.68_{\pm 0.30}$ |
| 10 | $\mathbf{40.61_{\pm 0.58}}$ | $\mathbf{79.91_{\pm 0.63}}$ | $23.25_{\pm 0.59}$ | $58.32_{\pm 0.73}$ |
| 20 | $40.42_{\pm 0.61}$ | $79.52_{\pm 0.49}$ | $23.35_{\pm 0.57}$ | $58.52_{\pm 0.44}$ |
| 50 | $40.25_{\pm 0.42}$ | $79.18_{\pm 0.36}$ | $23.44_{\pm 0.47}$ | $58.71_{\pm 0.32}$ |
| 100 | $40.19_{\pm 0.35}$ | $79.06_{\pm 0.42}$ | $\mathbf{23.48_{\pm 0.48}}$ | $\mathbf{58.77_{\pm 0.71}}$ |

**Meta-plasticity scheduler.** To validate the effectiveness of the meta-plasticity scheduler we introduce, we experiment with several different values for $\lambda$ in Equation 10 to observe its impact on overall performance. When $\lambda$ is set to 0, the corresponding parameters remain frozen, which is equivalent to a task-specific parameter freezing scheme. On the other hand, setting $\lambda$ to 1 effectively disables the meta-plasticity scheduler, meaning it has no effect. As shown in Table 4, freezing task-specific parameters proves to be effective, and the scheduler we introduce ($\lambda > 1$) further improves performance. The value of $\lambda$, as long as above 1, does not affect results much, while larger $\lambda$ leads to slightly better performance on more challenging tasks. This may be because as $\lambda$ increases, the meta-plasticity exhibits greater variability, making neurons more responsive to external inputs.

## 5 CONCLUSION

In this paper, we propose a framework that combines the continual weighted sparsity scheduler and the meta-plasticity scheduler to address the stability-plasticity trade-off in CL. The continual weighted sparsity scheduler iteratively prunes the network with progressively increasing sparsity over multiple rounds, leading to a more refined, task-specific group of neurons and connections, thereby preserving knowledge more effectively. Meanwhile, the meta-plasticity scheduler, inspired by biological meta-plasticity mechanisms, introduces connection-level and task-aware adjustments. This enables flexible, dynamic tuning of connections, supporting both knowledge retention and adaptation to new tasks. Experimental results demonstrate that our approach effectively balances stability and plasticity and outperforms other baselines. In the future, we aim to integrate dynamic network expansion into our framework to address challenges in real-world applications, which often involve a larger number of tasks, and potentially lack clear task boundaries.

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

## A DATASETS AND SETTINGS

We evaluate the effectiveness of our approach in two different types of CL scenarios: Class Incremental Learning (Class-IL) and Task Incremental Learning (Task-IL). In both settings, each task introduces a set number of new classes for the model to learn. A CL model learns these tasks sequentially while maintaining the ability to distinguish between all previously encountered classes. The key difference is that, in Task-IL, task labels are available during inference, making it a simpler scenario compared to Class-IL, where no such labels are provided.

### A.1 DATASET DETAILS

To evaluate the performance of our method in Task-IL and Class-IL scenarios, we employ three different datasets: CIFAR-10, CIFAR-100, and Tiny-ImageNet. The CIFAR-10 dataset consists of 60,000 $32 \times 32$ colored images in 10 classes, with 6000 images per class. There are 50,000 training images and 10,000 test images. CIFAR-100 is just like the CIFAR-10, except it has 100 classes containing 600 images each. There are 500 training images and 100 testing images per class. Tiny-ImageNet contains 100,000 images of 200 classes (500 for each class) downsized to $64 \times 64$ colored images. Each class has 400 training images, 50 validation images, and 50 test images.

## B  ADDITIONAL DETAILS ABOUT OUR METHOD

### B.1  PSEUDOCODE

We provide the detailed pseudo-code for our method as Algorithm 1.

---

**Algorithm 1:** Continual learning via continual weighted sparsity and meta-plasticity scheduling.

1 **Initialize**: working model $f_\theta$, data stream $D$, number of tasks $T$, target sparsity for each task $\Delta S$, total training steps for each task $N$.
2 $\mathcal{G} \leftarrow \{\}, D_m \leftarrow \{\}$;
3 **foreach** $t \in \{1, 2, \ldots, T\}$ **do**
4     Retrieve task data $D_t$ from $D$.
5     **foreach** $n \in \{1, 2, \ldots, N\}$ **do**
6         Update the target sparsity $S_t^n$ using Equation 4.     ▷ Sparsity scheduling
7         **foreach** batch of data $\mathcal{B}_t \subset D_t$ and $\mathcal{B}_m \subset D_m$ **do**
8             Update $f_\theta$ using Equation 1.
9         Prune neurons in $f_\theta$ using Equation 5.     ▷ Neuron selection
10         Drop connections using Equation 6.     ▷ Connection selection
11     Extract $g_t$, $\mathcal{G} \leftarrow \mathcal{G} \cup g_t$.
12     Update groups' meta-plasticity using Equation 10.     ▷ Meta-plasticity scheduling
13     Update $D_m$

---

### B.2  LAYER-WISE SPARSITY DISTRIBUTION

Given a target sparsity $S_{t,n}$ for the model, a uniform sparsity distribution is commonly used by setting the sparsity $S_{t,n}^{(l)}$ of each individual layer $l$ equal to the total sparsity $S_{t,n}$. However, applying the same level of sparsity to narrower layers may result in insufficient feature retention. To address this, we adopt the *Erdős-Rényi* (ER) method (Mocanu et al., 2018), which distributes the sparsity $S_t^{n,l}$ of each layer proportional to the term $\frac{d^{(l-1)}+d^{(l)}}{d^{(l-1)}\times d^{(l)}}$, where $d^{(l)}$ and $d^{(l-1)}$ are the numbers of neurons in layers $l$ and $l-1$, respectively. This method makes larger layers relatively more sparse than smaller ones. In the ER method, the input and output layers are relatively denser because they usually have fewer incoming or outgoing connections. This allows the network to better utilize the observations and learned representations at the highest layers in the network.

### B.3  LOSS FUNCTION

The loss function we used to update the working model $f_\theta$ here is introduced by Liang & Li (2024), they decouple the $\mathcal{L}_t$ in Equation 1 to two components:

$$\mathcal{L}_t = \mathbb{E}_{(x_i,y_i)\sim D_t}[\mathcal{L}_{\text{ce}}(f_\theta(x_i), y_i; t) + \mathcal{L}_n(f_\theta(x_i)))], \tag{11}$$

where $\mathcal{L}_{\text{ce}}(t)$ represents the loss on classes of the current task, and $\mathcal{L}_n$ represents the loss of classification of new/old class. Then two hyper-parameters are introduced to control the weight of the two different learning objectives:

$$\mathcal{L}'_t = \mathbb{E}_{(x_i,y_i)\sim D_t}[\beta_1 \mathcal{L}_{\text{ce}}(f_\theta(x_i), y_i; t) + \beta_2 \mathcal{L}_n(f_\theta(x_i))]. \tag{12}$$

Here, we adopt the optimal parameter combination used in the experiments from Liang & Li (2024), with $\beta_1 = 1$ and $\beta_2 = 0.1$.

## C  ADDITIONAL EXPERIMENTS

### C.1  STABILITY-PLASTICITY TRADE-OFF

A CL model is said to be stable if it can retain previously learned information, and plastic if it can effectively acquire new information. Following Sarfraz et al. (2022), let $a_{k,j} \in [0, 1]$ denote the

classification accuracy evaluated on the test set of the $j$-th task after incremental learning of the $k$-th task ($j \leq k$). The stability is evaluated by calculating the average performance across all preceding $T - 1$ tasks as:

$$\text{stability} = \frac{1}{T - 1} \sum_{j=1}^{T-1} a_{T,j}. \tag{13}$$

The models' plasticity can be accessed by computing the average performance of each task after its initial learning as:

$$\text{plasticity} = \frac{1}{T} \sum_{j=1}^{T} a_{j,j}. \tag{14}$$

Finally, the trade-off measure determines the optimal balance between the stability and the plasticity of the model. This measure is calculated as the harmonic mean of stability and plasticity:

$$\text{Trade-off} = \frac{2 \times \text{stability} \times \text{plasticity}}{\text{stability} + \text{plasticity}}. \tag{15}$$

### C.2 IMPLEMENTATION DETAILS

We run all the experiments on an NVIDIA GeForce RTX-3090Ti GPU. Our implementations are based on Ubuntu Linux 20.04 with Python 3.8. Additionally, we use ResNet-18 as the feature extractor for all of our investigations. We use the Adam optimizer with a learning rate of 0.001 at the beginning to train the model, and we use a batch size of 32 and train the model for 50 epochs for each task. Additionally, for all methods requiring a replay buffer, we consistently set the buffer size to 200 in all experiments, following the setup used in TriRE (Vijayan et al., 2023).

### C.3 ADDITIONAL EXPERIMENTAL RESULTS

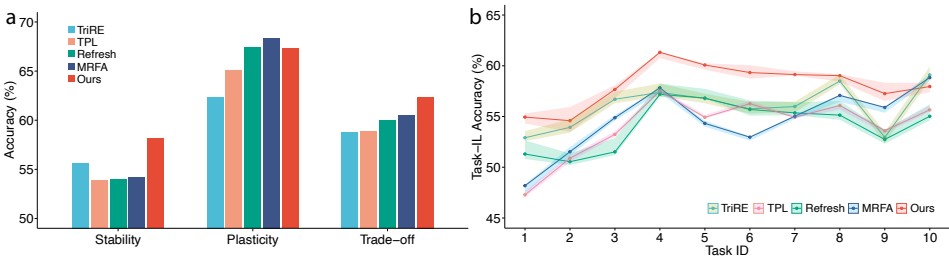

Figure 5: **a**. Stability-Plasticity trade-off for CL models trained on Tiny-ImageNet with 10 tasks. **b**. The comparison of our method against other representative baselines in terms of Task-IL accuracy on Tiny-ImageNet is divided into 10 tasks. The graph reports the average accuracy of individual tasks at the end of CL training in 5 runs with different seeds. The shaded area represents the error range determined by the maximum and minimum values.

**Stability-Plasticity trade-off.** We provide the trade-off between stability and plasticity achieved by our method, as well as the performance across all tasks after training on the Tiny-ImageNet with 10 tasks, with the results shown in Figure 5. Our method demonstrates the best stability while maintaining plasticity, which leads to the most favorable stability-plasticity trade-off. This explains why our approach achieves the best overall performance. Figure 5b provides additional insight, similar to the results in CIFAR-100 with 10 tasks. Our method significantly outperforms others on the earlier tasks.

**Performance on long sequences of tasks.** We provide the performance of all 20 tasks after training on the Tiny-ImageNet, as illustrated in Figure 6 and the stability-plasticity trade-off evaluation in Figure 7. Our method demonstrates superior stability by preserving the accuracy of the earlier tasks. Furthermore, it maintains relatively high accuracy and exhibits less fluctuation for newly added tasks compared to other methods, highlighting its plasticity.

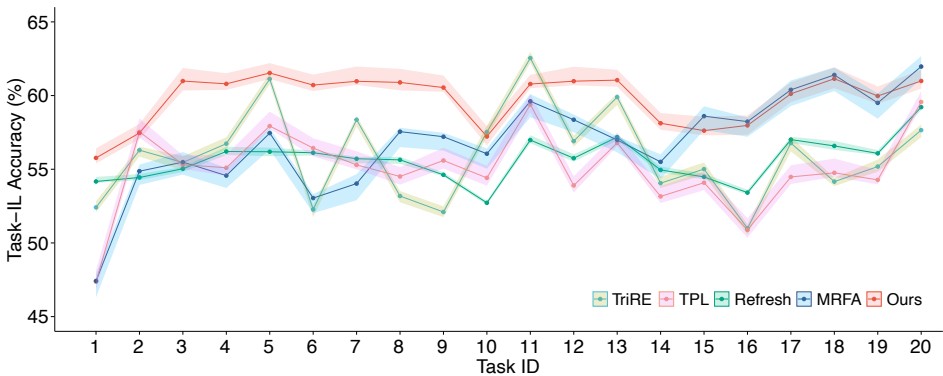

Figure 6: Comparison of our method against other representative baselines in terms of Task-IL accuracy on the Tiny-ImageNet dataset divided into 20 tasks. The graph reports the average accuracy of individual tasks at the end of CL training in 5 runs with different seeds. The shaded area represents the error range determined by the maximum and minimum values.

Table 5: Comparison of the average accuracy on ImageNet-1K.

| Methods | 10 Tasks | | 20 Tasks | |
|---|---|---|---|---|
| | Class-IL | Task-IL | Class-IL | Task-IL |
| TPL | $35.54_{\pm0.78}$ | $69.53_{\pm0.64}$ | $45.66_{\pm0.82}$ | $73.50_{\pm0.54}$ |
| Refresh | $36.30_{\pm0.65}$ | $69.04_{\pm0.56}$ | $46.61_{\pm0.58}$ | $72.09_{\pm0.46}$ |
| MRFA | $36.57_{\pm0.79}$ | $70.42_{\pm0.71}$ | $46.93_{\pm0.48}$ | $73.10_{\pm0.75}$ |
| **Ours** | $\mathbf{39.76_{\pm0.46}}$ | $\mathbf{74.94_{\pm0.47}}$ | $\mathbf{53.16_{\pm0.38}}$ | $\mathbf{78.49_{\pm0.42}}$ |

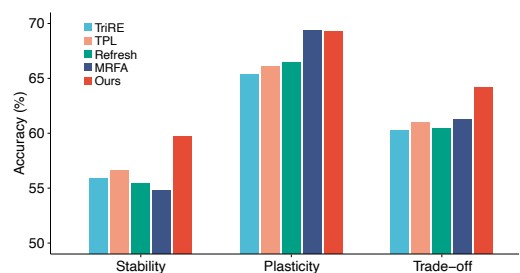

Figure 7: Stability-Plasticity trade-off for CL models trained on Tiny-ImageNet with 20 tasks.

**Performance on larger dataset ImageNet-1K.** To evaluate the scalability of our method to larger datasets, we have conducted an experiment on ImageNet-1K. For this larger dataset, we increased the buffer size to 2000 and compared our approach against three state-of-the-art baselines for both 10 tasks and 20 tasks settings. The results, shown in Table 5, demonstrate that our method continues to outperform the baselines.

**Comparison against the non-replay methods and pretrained-model based methods.** While replay-based methods have been widely used in CL, recent advancements have also introduced non-replay methods as promising alternatives. In addition, pretrained model-based approaches have emerged as a more advanced and broader frontier in CL. Here, we compare our method against these two types of baselines: 2 non-replay methods (FeTriL (Petit et al., 2023), DS-AL (Zhuang et al., 2024)), 2 pretrained-model based methods (Aper (Zhou et al., 2025), TPL (Lin et al., 2024)), using four datasets: CIFAR-10, CIFAR-100, Tiny-ImageNet, ImageNet-1K. The results, presented in Table 6 with the best result highlighted in bold, demonstrate our method consistently outperforms all baselines across all four datasets.

Table 6: Comparison of the overall performance across various CL scenarios.

| Methods | CIFAR-10 | | CIFAR-100 | | Tiny-ImageNet | | ImageNet-1K | |
|---|---|---|---|---|---|---|---|---|
| | Class-IL | Task-IL | Class-IL | Task-IL | Class-IL | Task-IL | Class-IL | Task-IL |
| FeTrIL | $69.45_{\pm0.81}$ | $90.37_{\pm0.69}$ | $35.20_{\pm0.70}$ | $72.59_{\pm0.57}$ | $20.39_{\pm1.03}$ | $54.01_{\pm0.66}$ | $35.12_{\pm0.63}$ | $69.86_{\pm0.86}$ |
| DS-AL | $71.59_{\pm0.47}$ | $92.05_{\pm0.43}$ | $38.40_{\pm0.32}$ | $76.93_{\pm0.40}$ | $22.01_{\pm0.59}$ | $54.97_{\pm0.53}$ | $36.36_{\pm0.66}$ | $70.04_{\pm0.53}$ |
| Aper | $73.25_{\pm0.99}$ | $93.21_{\pm0.78}$ | $37.48_{\pm0.39}$ | $77.47_{\pm0.62}$ | $21.59_{\pm0.46}$ | $54.42_{\pm0.67}$ | $37.26_{\pm0.84}$ | $69.37_{\pm0.79}$ |
| TPL | $70.06_{\pm0.47}$ | $92.33_{\pm0.32}$ | $36.90_{\pm0.42}$ | $76.53_{\pm0.27}$ | $20.06_{\pm0.77}$ | $54.20_{\pm0.51}$ | $35.54_{\pm0.78}$ | $69.53_{\pm0.64}$ |
| **Ours** | $\mathbf{75.31_{\pm0.71}}$ | $\mathbf{95.79_{\pm0.65}}$ | $\mathbf{40.61_{\pm0.58}}$ | $\mathbf{79.91_{\pm0.63}}$ | $\mathbf{23.25_{\pm0.59}}$ | $\mathbf{58.32_{\pm0.73}}$ | $\mathbf{39.76_{\pm0.46}}$ | $\mathbf{74.94_{\pm0.47}}$ |

**Comparison on different task-wise sparsity.** In the previous experiments, we allocated 15% of the neurons exclusively to each task, as this ratio was shown to be optimal according to the results from Graesser et al. (2022). To explore how this parameter affects performance, we conducted additional experiments comparing different task-specific sparsity ratios, as shown in Table 7. The results show that the performance for the 20% ratio is not as good as those for 15%. We believe that increasing the sparsity allocation may lead to more interference between tasks. On the other hand, when the ratio is set too small, there is a sharp decline in performance, which we attribute to the insufficient information retained by the selected neurons and connections.

Table 7: The average accuracy for different $\Delta S$ used in Equation 3.

| $\Delta S$ | CIFAR-100 | | Tiny-ImageNet | |
|---|---|---|---|---|
| | Class-IL | Task-IL | Class-IL | Task-IL |
| 20% | $40.18_{\pm 0.43}$ | $78.62_{\pm 0.44}$ | $22.59_{\pm 0.54}$ | $57.89_{\pm 0.57}$ |
| 15% | $\mathbf{40.61_{\pm 0.58}}$ | $\mathbf{79.91_{\pm 0.63}}$ | $\mathbf{23.25_{\pm 0.59}}$ | $\mathbf{58.32_{\pm 0.73}}$ |
| 10% | $38.49_{\pm 0.53}$ | $76.22_{\pm 0.45}$ | $21.68_{\pm 0.77}$ | $55.59_{\pm 0.73}$ |
| 5% | $37.22_{\pm 0.89}$ | $72.94_{\pm 1.01}$ | $21.66_{\pm 0.75}$ | $52.91_{\pm 0.91}$ |

**Comparison of different connection pruning methods.** There are two commonly used strategies to select the most important connections: (1) magnitude-based and (2) fisher information-based. The idea behind magnitude pruning is that small valued weights impact the network's output less and can be safely pruned without significantly affecting performance. Fisher information-based pruning evaluates the importance of connections based on their contributions to the Fisher information matrix. Connections with low contributions, indicating less relevance or importance, are pruned or set to zero. Wang et al. (2022a) proposed continual weighted importance (CWI), which considers not only the importance of weights within the current task but also the possibility of it being crucial for other tasks. Here, we extend the CWI by introducing an additional item $\|\frac{\partial \hat{\mathcal{L}}_{\text{new}}(D_t;\theta)}{\partial \omega}\|_1$, which consider the capacity of distinguishing the task boundary for the $\hat{\mathcal{L}}_{\text{new}}$ represents the cross entropy loss for new/old class distinction. To validate the effectiveness of the CWS we proposed, we compare it against the other three methods, with the result reported in Table 8. It can be observed that our proposed CWS can help improve the overall performance. Additionally, the improvement in Task-IL is relatively smaller compared to Class-IL, as the extension of CWI primarily enhances the model's ability to recognize task boundaries, a feature that is more crucial in the Class-IL setting.

Table 8: Comparison of the effect of various connection pruning methods used in Section 3.3 on different datasets.

| Methods | CIFAR-100 | | Tiny-ImageNet | |
|---|---|---|---|---|
| | Class-IL | Task-IL | Class-IL | Task-IL |
| Magnitude | $38.89_{\pm 0.71}$ | $77.29_{\pm 0.73}$ | $22.48_{\pm 0.79}$ | $56.69_{\pm 0.50}$ |
| Fisher-information | $37.26_{\pm 0.45}$ | $74.05_{\pm 0.51}$ | $21.54_{\pm 0.81}$ | $54.31_{\pm 0.78}$ |
| CWI | $39.88_{\pm 0.82}$ | $79.45_{\pm 0.83}$ | $22.86_{\pm 0.68}$ | $58.13_{\pm 0.56}$ |
| **Ours** | $\mathbf{40.61_{\pm 0.58}}$ | $\mathbf{79.91_{\pm 0.63}}$ | $\mathbf{23.25_{\pm 0.59}}$ | $\mathbf{58.32_{\pm 0.73}}$ |

**Task-wise and step-wise sparsity scheduling.** We provide the visualization of the target and real sparsity of the working model $f_\theta$ during training on CIFAR-100 with 10 tasks in Figure 8. As tasks are sequentially introduced, the total sparsity of the network gradually decreases while the sparsity gradually increases during the training process for each task. Furthermore, the real sparsity of the network at the end of each task does not match the target sparsity, due to some overlap between task-specific neuron groups.

**Comparison on different hyperparameters** $\alpha_1, \alpha_2, \beta_1$ **and** $\beta_2$**.** To examine the impact of the hyperparameters introduced in Equation 6 and Equation 12 on the performance, we conducted experiments comparing various hyperparameters, as shown in Tables 9 and 10. The results in Table 9 indicate that increasing $\alpha_2$ helps improve overall performance, as $\alpha_2$ ensures better retention of past

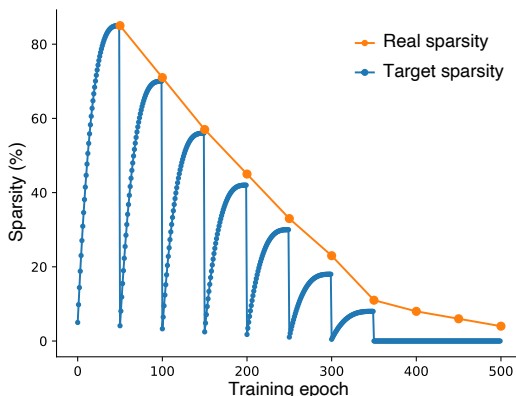

Figure 8: Visualization of sparsity scheduling results for CIFAR-100 with 10 tasks. Each task trains 50 epochs here. The target sparsity is calculated by Equation 4, while the real sparsity is calculated after training of each task by Equation 3.

Table 9: Comparison of different $\alpha_1$ and $\alpha_2$ used in Equation 6.

| $\alpha_1$ | $\alpha_2$ | CIFAR-100 | | Tiny-ImageNet | |
|---|---|---|---|---|---|
| | | Class-IL | Task-IL | Class-IL | Task-IL |
| 0.5 | 0.5 | $38.31_{\pm 0.71}$ | $77.01_{\pm 0.83}$ | $22.85_{\pm 0.37}$ | $57.18_{\pm 0.49}$ |
| 0.5 | 1 | $\mathbf{40.61_{\pm 0.58}}$ | $\mathbf{79.91_{\pm 0.63}}$ | $\mathbf{23.25_{\pm 0.59}}$ | $\mathbf{58.32_{\pm 0.73}}$ |
| 1 | 0.5 | $37.65_{\pm 0.80}$ | $76.80_{\pm 0.40}$ | $22.52_{\pm 0.91}$ | $57.04_{\pm 0.66}$ |
| 1 | 1 | $39.56_{\pm 0.62}$ | $77.25_{\pm 0.83}$ | $22.98_{\pm 0.69}$ | $57.33_{\pm 0.70}$ |

knowledge. In contrast, a larger $\alpha_1$ primarily enhances plasticity but may lead to forgetting previously learned knowledge, thereby reducing performance. In practical applications, it is advisable to keep $\alpha_2$ slightly greater than $\alpha_1$ to avoid excessively compromising the model's plasticity. Additionally, as shown in Table 10, the settings of $\beta_1$ and $\beta_2$ have a greater impact on Class-IL tasks compared to Task-IL tasks. We believe that Task-IL inherently has access to task IDs, reducing the model's reliance on distinguishing between new and old tasks. Furthermore, $\beta_1$ has a larger effect on performance than $\beta_2$, so it is recommended to ensure that $\beta_1 > \beta_2$ in practical applications.

Table 10: Comparison of different $\beta_1$ and $\beta_2$ used in Equation 12.

| $\beta_1$ | $\beta_2$ | CIFAR-100 | | Tiny-ImageNet | |
|---|---|---|---|---|---|
| | | Class-IL | Task-IL | Class-IL | Task-IL |
| 0.1 | 0.1 | $32.87_{\pm 0.71}$ | $75.38_{\pm 0.51}$ | $22.87_{\pm 0.75}$ | $56.64_{\pm 0.82}$ |
| 0.1 | 1 | $35.77_{\pm 0.52}$ | $77.52_{\pm 0.69}$ | $22.93_{\pm 0.74}$ | $57.44_{\pm 0.46}$ |
| 1 | 0.1 | $\mathbf{40.61_{\pm 0.58}}$ | $\mathbf{79.91_{\pm 0.63}}$ | $\mathbf{23.25_{\pm 0.59}}$ | $\mathbf{58.32_{\pm 0.73}}$ |
| 1 | 1 | $36.90_{\pm 0.85}$ | $78.16_{\pm 0.74}$ | $23.04_{\pm 0.91}$ | $57.18_{\pm 0.59}$ |

**Extension to vision transformers.** To further evaluate the generalizability of our method beyond convolutional architectures, we conducted additional experiments using the pretrained ViT/B-16 model. Specifically, we compared our approach with four baselines: replayed ViT (using the same replay buffer strategy as our method), L2P (Wang et al., 2022c), DualPrompt (Wang et al., 2022b), and EASE (Zhou et al., 2024). For fair comparison, buffer size and task split settings were kept consistent with those in other experiments. Since recent ViT-based continual learning studies mainly focus on the more challenging and realistic Class-IL scenario, we report Class-IL results for all methods on CIFAR-10, CIFAR-100, Tiny-ImageNet, and ImageNet-1K. As shown in Table 11, our method consistently outperforms both the replay-based and ViT-based baselines across all datasets. These results demonstrate that our framework is not limited to convolutional neural networks, but is also highly effective for transformer-based architectures.

Table 11: Comparison of different methods on ViT/B-16 across CIFAR-10, CIFAR-100, Tiny-ImageNet, and ImageNet-1K under Class-IL scenario.

| Methods | CIFAR-10 | CIFAR-100 | Tiny-ImageNet | ImageNet-1K |
|---|---|---|---|---|
| replayed ViT | $91.38_{\pm0.89}$ | $76.79_{\pm0.93}$ | $70.14_{\pm1.64}$ | $74.91_{\pm0.84}$ |
| L2P | $94.05_{\pm0.52}$ | $84.01_{\pm0.57}$ | $75.97_{\pm0.78}$ | $81.35_{\pm0.61}$ |
| DualPrompt | $94.87_{\pm0.27}$ | $86.51_{\pm0.38}$ | $77.78_{\pm0.49}$ | $81.97_{\pm0.31}$ |
| EASE | $95.09_{\pm0.51}$ | $92.35_{\pm0.48}$ | $78.55_{\pm0.62}$ | $82.83_{\pm0.59}$ |
| **Ours** | $\mathbf{95.65_{\pm0.33}}$ | $\mathbf{92.98_{\pm0.40}}$ | $\mathbf{80.92_{\pm0.58}}$ | $\mathbf{84.52_{\pm0.43}}$ |

## C.4 COMPUTATIONAL COST

We compare the training times of various continual learning (CL) models considered in this work, selecting one representative method from each family of CL approaches. Table 12 presents the training times required to learn a total of 10 tasks, with each task trained for 50 epochs on the CIFAR-100 dataset using an NVIDIA GeForce RTX-3090Ti and a buffer size of 200. The training time comparison indicates that our method requires longer durations due to the incorporation of fine-grained, neuron-level computations during training, especially the phase of sparsity scheduling.

Table 12: Comparison of the training time across various methods.

| Methods | DER++ | TriRE | Ours | |
|---|---|---|---|---|
| | | | Continual weighted sparsity scheduler | Meta-plasticity scheduler |
| Training time (hours) | 4.03 | 4.61 | 4.85 | 0.79 |

**Sparsity scheduling frequency.** During the training process for each task, we now do the sparsity scheduling for each training epoch, which is time-consuming and computing-consuming. A typical solution is to use a lazy update strategy, such as updating the sparsity periodically. Here, we extend the sparsity scheduling to a periodical version with the period denoted as $\Delta T$, then we have:

$$S_{t,n} = S_{t,N} - S_{t,N}(1 - \frac{n}{N})^3, \qquad n = \Delta T, 2\Delta T, \ldots, N. \tag{16}$$

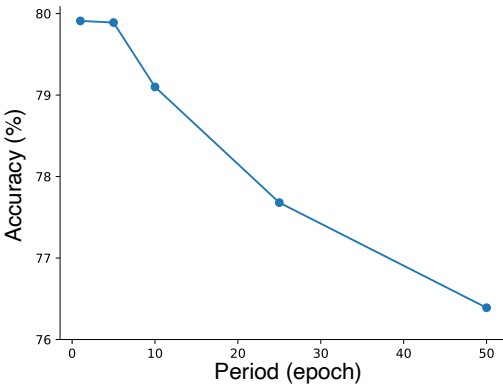

Figure 9: The average accuracy on CIFAR-100 (10 tasks) for different $\Delta T$ used in Equation 16. We set $\Delta T = 1, 5, 10, 25, 50$ here, with each task training for 50 epochs.

We compare the impact of different $\Delta T$ values on the average accuracy of the CIFAR-100 with 10 tasks (Figure 9). When $\Delta T$ is set to the total number of training epochs, the method effectively reduces to static sparse training. As illustrated in Figure 9, changing the update interval from 1 to 5 has minimal impact on performance. However, when updates become too infrequent, such as only occurring once per task, there is a noticeable drop in performance. Additionally, in Table 13, we

present a detailed comparison of the time consumption and accuracy performance for $\Delta T = 1, 2, 5$. As observed, increasing $\Delta T$ significantly reduces training time, albeit with a slight drop in accuracy. However, this trade-off is acceptable from a practical standpoint. Based on these results, to balance computational efficiency and performance, we recommend using an update interval of 5 epochs.

Table 13: Comparison of training time and performance for $\Delta T = 1, 2, 5$ used in Equation 16.

| $\Delta T$ | Training time (hours) | CIFAR-100 | | Tiny-ImageNet | |
|---|---|---|---|---|---|
| | | Class-IL | Task-IL | Class-IL | Task-IL |
| 1 | 5.64 | $\mathbf{40.61}_{\pm 0.58}$ | $\mathbf{79.91}_{\pm 0.63}$ | $\mathbf{23.25}_{\pm 0.59}$ | $\mathbf{58.32}_{\pm 0.73}$ |
| 2 | 4.67 | $40.54_{\pm 0.64}$ | $79.89_{\pm 0.52}$ | $23.15_{\pm 0.85}$ | $58.21_{\pm 0.65}$ |
| 5 | 4.11 | $40.41_{\pm 0.57}$ | $79.85_{\pm 0.78}$ | $22.99_{\pm 0.47}$ | $58.07_{\pm 0.69}$ |

