# OpenReview forum: "Continual Learning via Continual Weighted Sparsity and Meta-Plasticity Scheduling"
_ICLR.cc/2026/Conference — Submitted to ICLR 2026_

### Official Review · Reviewer_HHV3 · 2025-10-20

**Soundness:** 2
**Presentation:** 3
**Contribution:** 2
**Rating:** 4
**Confidence:** 4

**Summary:**

This paper presents a continual learning approach using the parameter isolation technique and regularization through learning rate tuning. It claims to address the stability-plasticity dilemma in CL.

**Strengths:**

1) This paper is well-written;
2) although the idea is common, the algorithm instantiation is new.

**Weaknesses:**

1) contribution of this paper should be elaborated more in section 1 of this paper. currently, it is hard for readers to grasp the key contribution of this paper.

2) why is memory still required in your case? If the parameter isolation or regularization strategy work properly, the memory is not required at all.

3) Controlling the learning rates for addressing the catastrophic forgetting problem is not new in CL. Authors should review these works in the related works section.

Overcoming Catastrophic Forgetting by Neuron-level Plasticity Control

Continual learning via inter-task synaptic mapping

4) datasets used are not sufficiently complex. I suggest add experiments with ImageNet-R or CUB-200

5) algorithm's complexity should be analyzed using a big O notation

**Questions:**

1) contribution of this paper should be elaborated more in section 1 of this paper. currently, it is hard for readers to grasp the key contribution of this paper.

2) why is memory still required in your case? If the parameter isolation or regularization strategy work properly, the memory is not required at all.

3) Controlling the learning rates for addressing the catastrophic forgetting problem is not new in CL. Authors should review these works in the related works section.

Overcoming Catastrophic Forgetting by Neuron-level Plasticity Control

Continual learning via inter-task synaptic mapping

4) datasets used are not sufficiently complex. I suggest add experiments with ImageNet-R or CUB-200

5) algorithm's complexity should be analyzed using a big O notation

---

### Official Review · Reviewer_XYUV · 2025-10-27

**Soundness:** 2
**Presentation:** 2
**Contribution:** 2
**Rating:** 2
**Confidence:** 4

**Summary:**

This paper presents a novel framework for addressing the stability-plasticity trade-off in Continual Learning (CL). The proposed approach integrates two components: the Continual Weighted Sparsity Scheduler and the Meta-Plasticity Scheduler. The former utilizes a dynamic sparse training technique to refine task-specific groups of neurons, guiding iterative pruning processes over several rounds. The latter draws inspiration from biological models to adjust learning rates dynamically based on a sensitive score function, ensuring balanced retention of old knowledge and acquisition of new skills. The experimental results demonstrate that this method consistently surpasses existing state-of-the-art approaches across several standard datasets, such as CIFAR-10, CIFAR-100, Tiny-ImageNet, and ImageNet-1K. The framework achieves this by optimizing both stability and plasticity.

**Strengths:**

1. **Novel Approach**: The combination of continual weighted sparsity and meta-plasticity scheduling seems to be interesting and novel, providing a sophisticated mechanism to balance stability and plasticity.

2. **Relatively Thorough Evaluation**: The authors evaluated their method against several baselines combined with ablations to validate their approach.

3. **Biologically Inspired Design**: Drawing inspiration from the brain’s meta-plasticity mechanisms adds theoretical depth and potentially broadens applicability to more dynamic and complex environments.

**Weaknesses:**

1. **Lack of Clarity:** The explanation or notation for some definitions and concepts is missing, e.g.,
    *  In eq.(6), the gradient is w.r.t. $w$, but the strict definition of $w$ is missing, and the losses do not contain a parameter for $w$.
    * Again in eq.(6), there is no definition for $\hat{\mathcal{L}}_{\text{new}}(D_t;\theta)$.
    * In experiments, the definition of stability, plasticity, and their trade-off are missing in the main paper. Please move them from the Appendix to the main paper.

2. **Lack of Related Work Discussion:** The paper only discussed replay-based and parameter isolation methods, while regularization techniques and meta-learning-based approaches are not discussed, especially those explicitly studied dynamic network expansion, stability and plasticity trade-off, such as [1-8].

3. **Complex Implementation**: The continual weighted sparsity scheduler and meta-plasticity scheduling require careful tuning of numerous parameters, which might demand substantial computational resources and expertise for effective deployment.


4. **Sparsity Allocation**: The paper mentions performance degradation when varying task-specific sparsity ratios, indicating potential limitations in adjusting these parameters.

5. **Limited Task Length:** Even though the authors claim a long task sequence, only $20$ tasks were considered. Previous works have considered super long tasks [2, 4, 8].

[1] Chelsea Finn, Aravind Rajeswaran, Sham Kakade, and Sergey Levine. Online meta-learning. In International Conference on Machine Learning, pages 1920–1930. PMLR, 2019.

[2] Massimo Caccia, Pau Rodriguez, Oleksiy Ostapenko, Fabrice Normandin, Min Lin, Lucas Page-Caccia, Issam Hadj Laradji, Irina Rish, Alexandre Lacoste, David Vázquez, et al. Online fast adaptation and knowledge accumulation (osaka): a new approach to continual learning. Advances in Neural Information Processing Systems, 33:16532–16545, 2020.

[3] Giulia Denevi, Carlo Ciliberto, Riccardo Grazzi, and Massimiliano Pontil. Learning-to-learn stochastic gradient descent with biased regularization. In International Conference on Machine Learning, pages 1566–1575. PMLR, 2019.

[4] Qi Chen, Changjian Shui, Ligong Han, and Mario Marchand. On the stability-plasticity dilemma in continual meta-learning: Theory and algorithm. Advances in Neural Information Processing Systems, 36:27414–27468, 2023.

[5] Maria-Florina Balcan, Mikhail Khodak, and Ameet Talwalkar. Provable guarantees for gradient-based meta-learning. In International Conference on Machine Learning, pages 424–433. PMLR, 2019.

[6] Mikhail Khodak, Maria-Florina Balcan, and Ameet Talwalkar. Adaptive gradient-based meta-learning methods. arXiv preprint arXiv:1906.02717, 2019.

[7] Qiang Zhang, Jinyuan Fang, Zaiqiao Meng, Shangsong Liang, and Emine Yilmaz. Variational continual Bayesian meta-learning. Advances in Neural Information Processing Systems, 34: 24556–24568, 2021.

[8] Zhenyi Wang, Li Shen, Tiehang Duan, Donglin Zhan, Le Fang, and Mingchen Gao. Learning to learn and remember super long multi-domain task sequence. In Proceedings of the IEEE/CVF
Conference on Computer Vision and Pattern Recognition, pages 7982–7992, 2022.

**Questions:**

1.  **Unclear where the Improvements are from:**
    * The method filters the neurons into groups, and neurons with lower activation values are pruned. What's the difference compared to normal NN updates, where neurons with lower activation will be changed less (a soft regularization version)?
    * The sparsity seems to be from the $L_1$ regularization in eq. (6). What's the effect of the $\alpha$s? As this already puts a constraint on sparsity, why do you need the above Sparsity scheduling?
    * How does the memory buffer affect the results? Is it fairly set for all baselines?

2. **Dynamic Network Expansion**: The authors suggest integrating dynamic network expansion in future work. How might this change the current framework's effectiveness on tasks with blurred boundaries or real-world applications with numerous tasks?

3. **Sensitivity to Initial Conditions**: Given the different initializations mentioned, how sensitive is the model to such variations in practice, and what implications might this have on its deployment in diverse real-world scenarios?

4. **Hyper-parameter**: How are the hyper-parameters tuned?

I am happy to increase the score if most of my concerns are well addressed.

---

### Official Review · Reviewer_ZmNd · 2025-10-29

**Soundness:** 3
**Presentation:** 2
**Contribution:** 3
**Rating:** 4
**Confidence:** 5

**Summary:**

This paper introduces a novel framework for addressing the stability-plasticity dilemma in Continual Learning (CL). To maintain model stability, it proposes a dynamic sparse training scheme that iteratively prunes redundant parameters when learning a new task, preserving past knowledge while resulting in a more refined parameter group. Moreover, to enhance model plasticity, it computes the parameter sensitivity of each group and assigns dynamic learning rates to different parameters according to sensitivity scores. Extensive experiments validate the effectiveness of each component of the proposed method.

**Strengths:**

The overall framework is well-motivated. This paper proposes a dynamic pruning strategy to preserve sparse but important parameters for old tasks, achieving better stability than post-training pruning algorithms. It also introduces a dynamic learning rate adjustment scheme rather than directly freezing task-specific parameter groups, resulting in better model plasticity.
The experiments are comprehensive. The proposed framework demonstrated both improvement in stability and plasticity. The paper is well-written and easy to follow.

**Weaknesses:**

The contributions of the sparse pruning scheme appear to be moderate. Network pruning and parameter isolation have been extensively studied in the context of continual learning [1, 2, 3]. The proposed scheduling strategy seems to be a variant of these prior approaches, transforming static post-training pruning into a dynamic and iterative process. It appears more like a heuristic trick rather than a fundamentally advanced framework. The authors are encouraged to conduct a deeper analysis of related works and provide a more thorough explanation of the underlying mechanism behind the proposed iterative pruning strategy.
The proposed meta-plasticity strategy modulates the learning rate based on group-wise sensitivity, which appears conceptually similar to other parameter sensitivity–based regularization methods such as EWC [4] and UPGD [5]. The authors are encouraged to further clarify the distinctions and advantages of the proposed approach over these prior works. For example, how the group-wise sensitivity mechanism improves upon the parameter-wise sensitivity adopted in existing methods.
The authors are suggested to provide details of the memory buffer size of the proposed method and existing SOTAs in Table 1.
[1] Wang Z, Zhan Z, Gong Y, et al. Sparcl: Sparse continual learning on the edge[J]. Advances in Neural Information Processing Systems, 2022, 35: 20366-20380.
[2] Kang H, Mina R J L, Madjid S R H, et al. Forget-free continual learning with winning subnetworks[C]//International conference on machine learning. PMLR, 2022: 10734-10750.
[3] Gao Q, Shan X, Zhang Y, et al. Enhancing knowledge transfer for task incremental learning with data-free subnetwork[J]. Advances in Neural Information Processing Systems, 2023, 36: 68471-68484.
[4] Kirkpatrick J, Pascanu R, Rabinowitz N, et al. Overcoming catastrophic forgetting in neural networks[J]. Proceedings of the national academy of sciences, 2017, 114(13): 3521-3526.
[5] Elsayed M, Mahmood A R. Addressing loss of plasticity and catastrophic forgetting in continual learning[J]. arXiv preprint arXiv:2404.00781, 2024. (ICLR 2024)

**Questions:**

See Weaknesses.

---

> ### Comment · Reviewer_ZmNd · 2025-11-26
>
> I keep my score as the authors do not give any response.

---

### Official Review · Reviewer_42VU · 2025-11-05

**Soundness:** 3
**Presentation:** 1
**Contribution:** 2
**Rating:** 4
**Confidence:** 4

**Summary:**

The paper proposes a new framework for continual learning that addresses the stability–plasticity dilemma. The method introduces two key components: a continual weighted sparsity scheduler, which iteratively prunes neurons and connections to form refined, task-specific subnetworks, and a meta-plasticity scheduler, which dynamically adjusts learning rates based on sensitivity scores inspired by biological mechanisms. Together, these techniques enhance stability through parameter isolation and preserve adaptability by regulating neural plasticity. Experiments on several benchmarks (CIFAR-10, CIFAR-100, Tiny-ImageNet, and ImageNet-1K) demonstrate that the proposed approach consistently outperforms existing replay-based and parameter-isolation methods in both class-incremental and task-incremental settings.

**Strengths:**

The method is technically sound. Results show consistent and significant improvements over strong baselines, supported by ablation studies and visual analyses that validate the method’s effectiveness.

**Weaknesses:**

1. The paper lacks clarity in presentation. The methodology is confusing, with several important details insufficiently explained.
2. The main contributions is introducing the new loss term $L_{\text{new}}$ in CWS and the meta-plasticity scheduler. But these contributions are relatively incremental. The paper does not clearly justify the motivation or necessity of these components.
3. The design and role of $L_{\text{new}}$ are unclear. It appears to introduce an auxiliary classification objective and potentially additional parameters, yet this loss is not explicitly optimized or detailed in the algorithm.
4. The rationale for defining the sensitivity score $SS$ based on $C_e$ is not well-motivated. The paper does not sufficiently explain how this formulation helps mitigate catastrophic forgetting since it does not freeze any parameters.
5. Experimental coverage is incomplete. Some relevant baselines (e.g., PackNet) are missing, and certain implementation details or results are not reported. More experiments beyond accuracy, i.e., sparsity, plasticity, etc. would be more helpful to demonstrate the effectiveness of the proposed method.

**Questions:**

1. Does the proposed method rely on class meta-information (e.g., class names or descriptions) during training?
2. How does the sensitivity score $SS$  evolve across tasks as training progresses?
3. How does the meta-plasticity scheduler specifically prevent catastrophic forgetting? Since learning rates are reduced only after significant parameter changes, could forgetting have already occurred before learning rates reduction?
4. Are any parameters frozen during training, or are all non-pruned parameters continuously updated across tasks?

---

### Meta-Review · Area_Chair_vrHP · 2026-01-07

**Summary:**

Contributions appear incremental and weakly motivated.
The paper claims its main contributions are a new loss term in CWS and a meta-plasticity scheduler, but reviewers find these additions relatively minor and not well justified. The motivation for these components is unclear, and the paper does not clearly explain the key novelty, making it hard for readers to grasp the central contribution.

Sensitivity score definition is poorly justified.
The rationale for defining the sensitivity score from gradient-based quantities is not well motivated. Reviewers question how the formulation mitigates catastrophic forgetting since it does not explicitly freeze parameters, and they ask how sensitivity evolves across tasks during training.

Timing concerns on Meta-plasticity scheduler:
Reviewers question whether reducing learning rates only after “significant parameter changes” can truly prevent forgetting, since forgetting may have already occurred by the time the learning rate is reduced. They request a clearer mechanistic explanation of how the scheduler protects prior knowledge and why group-wise sensitivity is preferable to parameter-wise approaches.

Reviewers ask why a memory buffer is still required if parameter isolation or regularization is supposed to work, and whether the method relies on class meta-information (class names/descriptions) during training.

**Reviewer Concerns:**

The authors do not provide response. The reviewer concerns remained after rebuttal.

**Reviewer Scores:**

No change in score.

---

### Decision · Program_Chairs · 2026-01-26

Reject